# Scaffold Structural Microenvironmental Cues to Guide Tissue Regeneration in Bone Tissue Applications

**DOI:** 10.3390/nano8110960

**Published:** 2018-11-21

**Authors:** Xuening Chen, Hongyuan Fan, Xiaowei Deng, Lina Wu, Tao Yi, Linxia Gu, Changchun Zhou, Yujiang Fan, Xingdong Zhang

**Affiliations:** 1National Engineering Research Center for Biomaterials, Sichuan University, Chengdu 610064, China; xchen6@scu.edu.cn (X.C.); wulina406410492@163.com (L.W.); yujiang.fan@163.com (Y.F.); zhangxd@scu.edu.cn (X.Z.); 2Scholl of Manufacturing Science and Engineering, Sichuan University, Chengdu 610065, China; fanhy@scu.edu.cn (H.F.); ws1tao@163.com (T.Y.); 3Department of Civil Engineering, The University of Hongkong, Pokfulam, Hongkong 999077, China; xwdeng@hku.hk; 4Department of Mechanical and Materials Engineering, University of Nebraska-Lincoln, Lincoln, NE 68588-0526, USA

**Keywords:** structural microenvironmental cues, porosity, grain size, surface topography, bone tissue regeneration

## Abstract

In the process of bone regeneration, new bone formation is largely affected by physico-chemical cues in the surrounding microenvironment. Tissue cells reside in a complex scaffold physiological microenvironment. The scaffold should provide certain circumstance full of structural cues to enhance multipotent mesenchymal stem cell (MSC) differentiation, osteoblast growth, extracellular matrix (ECM) deposition, and subsequent new bone formation. This article reviewed advances in fabrication technology that enable the creation of biomaterials with well-defined pore structure and surface topography, which can be sensed by host tissue cells (esp., stem cells) and subsequently determine cell fates during differentiation. Three important cues, including scaffold pore structure (i.e., porosity and pore size), grain size, and surface topography were studied. These findings improve our understanding of how the mechanism scaffold microenvironmental cues guide bone tissue regeneration.

## 1. Natural Bone Structure

Bones are important structural components of vertebrates, which play essential roles in providing mechanical support for locomotion, protecting vital organs, and regulating the metabolism of calcium and phosphorus. The lifelong execution of these bio-functions requires healthy bone tissues. In nature, bone is a type of dense connective tissue and shows strong mechanical strength. Bone tissue is a natural organic–inorganic composite, which shows extremely strong and elastic properties (Compressive strength of compact bone: 89–164 MPa; Elastic modulus of human bone: ~19.6 GPa). Living cells are embedded in a mineralized bone extracellular matrix (ECM), which consists of approximately 30 wt% organic matrix (approximately 95% type I collagen and ~5% other non-collagenous proteins) and ~70 wt% inorganic nanocrystallites (mainly carbonated hydroxyapatite crystals) [1]. There are two main types of bones that are called as cancellous (or trabecular) bone and cortical (or compact) bone. The cancellous bone has a porous structure with a network of trabeculae (porosity of 40–95%), while, the denser cortical bone (porosity of 5–25%) is composed of longitudinally oriented cylindrical elements, known as osteons [2,3]. A single osteon consists of concentric lamellae. Collagen fibers are parallel to each other in each lamella, but run in the opposite angle between the adjacent lamellae. The rod-like nano-crystalline inorganic HA particles (20–80 nm long, 2–5 nm thick) are embedded into collagen fibers to increase the rigidity of bone [4,5]. The hierarchical anatomy structures of bone tissues are illustrated in Figure 1. The chemical composition and hierarchical structure of bone tissues render them with unique biological properties and relatively high compressive strength.

## 2. Bone Defects and Bone Tissue Regeneration

Bone tissue defects are increasingly becoming the majority of clinical cases in orthopedics. Every year, millions of people worldwide suffer from bone tissue defects due to trauma, skeletal diseases, tumor resections, osteoporosis-related fractures, congenital bone malformations, aging and so on. Nowadays, there is an urgent need for bone regeneration [6]. Currently, autograft is considered as the gold standard. However, it suffers from limited resource and donor-site morbidity due to potential infection and haematoma. Allograft may suffer from immunological rejection and some ethic troubles [7]. Artificial synthetic grafts (e.g., titanium, calcium phosphate ceramics) provide alternatives for orthopaedic therapy [8]. An “ideal” scaffold should possess excellent biocompatibility, osteoconduction, and even osteoinduction, which means the capability of recruiting and inducing multipotent mesenchymal stem cells (MSCs) to differentiate into mature osteoblasts (bone-forming cells) [9,10]. Therefore, this review focuses on exploring how the structural microenvironmental cues of scaffolds direct the osteogenic differentiation of stem cells (MSCs), in order to provide guidance in design and development of promising scaffolds for bone regeneration.

## 3. Effects of Scaffold Structural Cues on Biological Responses

Upon implantation into body, orthopedic implant directly contacts with host tissues. Scaffold structural properties play a critical role in regulating cellular responses, including cell adherent, spreading, proliferation, and differentiation [11,12,13]. Recently, advances in fabrication technology enable to create biomaterials with well-defined pore structure and surface topography, which can be sensed by host tissue cells (esp., stem cells) and subsequently determine cell fates during differentiation. There are three major cues such as scaffold pore structure (i.e., porosity and pore size), grain size, and surface topography (as shown in Figure 2). The porosity of scaffolds not only provides space for the cell settlement and growth, but also ensures the transport of nutrients and metabolites. Too large pore sizes are not conducive to cell habitation, and too small pore sizes are not good for cell migration and proliferation. Moreover, the grain size of scaffolds affects protein adsorption. It has been reported that small grain size (such as nanocrystal) provides more adsorption sites, which are more beneficial to cell adhesion and proliferation [14,15]. The surface topography of scaffolds directly participates in biomaterial-tissue interface, and their roughness affects cell adhesion and crawling.

### 3.1. Porosity and Pore Size Cues

Scaffolds in tissue engineering ought to act as the temporary extracellular matrix to provide supports for cells and guide cell differentiation [16,17,18]. Porous bioactive materials are designed to mimic the properties of in vivo environment [19]. Their porosity and pore size are crucial in determining biological functions. Generally speaking, scaffolds with high porosity result in good interaction with the surrounding tissues and promote ingrowth of bone cells in vivo, while, high porosity also causes the diminished mechanical properties of scaffolds [20]. Therefore, the porosity of scaffolds must be designed to satisfy both their mechanical features and biological performance.

It is well known that a scaffold with porous structure can favor cell ingrowth and allow long-term stable fixation with the surrounding host tissues. Porous structure usually refers to porosity, pore size, surface area, connectivity, distortion degree of connected channels, and so on. An appropriate porous structure of scaffold plays a crucial role in achieving an optimal osteogenic effect [21,22], as high porosity and open structure are necessary for the ingrowth of bone tissues and blood vessels, and also ensure bone oxygenation [23,24,25]. The minimal pore size for bioactive porous material was reported as approximately 100 μm, which was appropriate for cell migration and nutrient transport [23]. Previous studies also found that pores with size above 200 μm could promote new bone formation and vascularization [24]. It is generally believed that high porosities (>80%) are optimal for new bone tissue regeneration, and macroporosity with pore sizes of 100–300 μm is beneficial to waste removal and nutrient supply [26,27]. Literature also suggested that small pores with sizes in a range of 50 to 100 μm were better for inducing endochondral ossification (i.e., osteochondral formation prior to osteogenesis), while, large pores (100–300 μm) facilitated vascularization and induced intramembranous ossification (i.e., bone formation without preceding cartilage formation) [26,28]. These findings have demonstrated that bioactive scaffold should exhibit proper porous structure with suitable porosity and pore size [29,30,31]. Moreover, some researchers have pointed out that cell ingrowth depends not only on the size of apertures, but also on the degree of connectivity and the size of channels. There is an urgent need to systematically investigate the roles of porosity and pore size in osteogenic outcome, which will help us to design and fabricate orthopedic implants with the desired clinical performance [32,33].

### 3.2. Grain Size Cues

Numerous studies have demonstrated that scaffold structural dimension (grain size) can significantly affect the behaviors of osteoprogenitors (e.g., MSCs) and osteoblasts. The grain size will change the specific surface area of biomaterials and affect cell adhesion, proliferation, and differentiation, which play an important role in bone tissue regeneration. Nowadays, CaP-based bioceramics with a similar chemical composition but different grain sizes in a range of nano-scale (≦100 nm) to submicron-scale (100 nm–1 μm) and micron-scale (≧1 μm) have been fabricated, and the effects of grain sizes on biological responses have been extensively investigated. Our group produced two kinds of porous hydroxyapatite (HA)/tricalcium phosphate (β-TCP) ceramics via H_2_O_2_ foaming by sintering at 1100 °C (HT11) and 1200 °C (HT12), respectively. These ceramics exhibited the similar phase composition and macro-porous structures, but HT11 had significantly smaller grain size than HT12. The dog intramuscular implantation experiment showed that HT11 ceramics with smaller grain size induced earlier bone formation and larger new bone area than HT12 [34]. Similar results have been found in other studies [35,36,37,38,39,40], suggesting that changes in sintering temperatures can modulate micro-structure of CaP-based bioceramics, as the increment of sintering temperature gradually increases crystal grain size, but exerts few effects on their chemical composition and macro-porous structures.

Some researchers also compared the biological responses of CaP bioceramics with grain sizes in submicron-scale and micron-scale [41,42,43]. Two kinds of β-TCP bioceramics (abbr., TCPs and TCPb) with equivalent chemical composition but varied grain sizes were fabricated by using different TCP powders as starting materials and adjusting reaction conditions. Compared to TCPb that exhibited micro-grains in size of 3–4 μm, TCPs with a grain size below 1 μm could promote osteogenic differentiation of human BMSCs by increasing alkaline phosphatase (ALP) activity and up-regulating expression of osteogenic specific genes (i.e., osteopontin and osteocalcin) in vitro. Upon implantation into dog dorsal muscles, TCPs with submicron-scale grains could induce ectopic bone formation, while, no bone tissue was found in TCPb with micron-sized surface architecture [41]. Further studies found that TCPs might induce osteoblastic differentiation of BMSCs by enhancing osteoclastic differentiation and promoting the secretion of pro-osteogenic factors in osteoclasts [42]. Whereas, TCPs with liposome-encapsulated clodronate (TCPs + LipClod) could not induce any subcutaneous bone formation, as LipClod depleted the osteoclast progenitors–monocytes/macrophages [43]. These findings have demonstrated that as compared to those with micro-grains, CaP ceramics with submicron-scale surface architecture exhibit superior osteoinductivity, which may be attributed to the enhanced osteoclastogenesis induced by submicro-grains.

Moreover, natural bone is composed of nanosized and nanocrystalline hydroxyapatites [44,45], and bone biomineralization process indicates that nanosized grains play an important role in the formation of hard tissues. Therefore, nanocrystalline forms of calcium phosphates have great potential for bone tissue regeneration [46,47,48,49,50]. Previous researchers found that nanocrystalline calcium deficient hydroxyapatite (CDHA) [51,52] and β-tricalcium phosphate (β-TCP) [53] exhibited the enhanced densification and improved sinterability due to their greater surface areas. Hao et al. [54] reported that 67 nm nanosized HA had a significantly higher surface roughness than 180 nm submicron-sized HA, and the contact angles of nanosized HA and conventional micron-sized HA were 6.1 and 11.51, respectively. Pielichowska et al. [55] demonstrated that nanosized HA had 11% more protein adsorption per one square centimeter than conventional micron-sized HA. Furthermore, extensive studies have demonstrated that nanosized HA ceramics show better bioactivity than the ones with coarser micron-sized crystals [56,57]. For instance, some researchers found [58,59] that titanium with nanocrystalline HA coatings could dramatically increase osteoblast adhesion, as compared with the one with traditionally used plasma-sprayed micron-sized HA coatings. Kim et al. [60] also reported that a larger number of osteoblasts attached onto nanosized HA/gelatin biocomposites than the micrometer-sized analogues. In addition, nanophase HA could promote proliferation and induce osteogenic differentiation of periodontal ligament cells compared to a dense HA bioceramic [61]. Our group also fabricated HA and biphasic calcium phosphate (BCP) nanoceramics with grain size as 115 ± 21 nm and 86 ± 20 nm, respectively [62]. Compared to traditional HA and BCP ceramics with submicron-sized grains (~700 nm), nanoceramics could promote osteoblastic differentiation in vitro by up-regulating osteogenic marker genes (e.g., BMP-2 and Cbfa1/Runx2) and increasing ALP activity, and induce more ectopic bone formation in vivo [47,62]. To sum up, these findings have demonstrated that compared to their microstructured counterparts, nanostructured biomaterials offer improved biological performances, which may be attributed to their high specific surface area, large surface-to-volume ratio, abundant surface defects, and unusual chemical synergistic effects.

### 3.3. Surface Topography of Scaffolds

As the surface of scaffold is directly in contact with host living tissues, the effects of scaffold surface on biological performances have been extensively investigated. Surface topographic cues (e.g., roughness, stiffness, and texture) play important roles in regulating cell responses and determining cell fates around the implants [63,64].

#### 3.3.1. Microscale Surface Topography

Researchers have created micropatterns of well-defined geometric features on polymeric substrates to control shape and spreading degree of single stem cells (MSC confinement) via lithography and microcontact printing. These micropatterns are composed of adhesion-promoting fibronectin to generate defined adhesive islands for single cell adhesion, and otherwise the remaining areas of substrates prevent protein adsorption and cell attachment. McBeath et al. [65] found that small fibronectin islands (cell-confining, 1024 μm^2^) favored adipogenesis, while, large islands (pro-spreading, 10,000 μm^2^) promoted osteogenesis. It suggested that micropattern sizes (single cell sizes) might have a marked impact on cell differentiation, which was closely correlated with a RhoA-ROCKmediated cytoskeletal tension. Mrksich’s group [66,67] fabricated micropatterns (adhesive islands) with the same area (1000 μm^2^) but diverse shapes, including rectangles with varying aspect ratios and pentagonal shapes with different subcellular curvature. It revealed that local shape cues (e.g., subcellular curvature) that increased myosin contractility could promote an osteogenic outcome instead of an adipogenic one through mitogen-activated protein kinase (MAPK) and Wnt-related signaling pathways. These findings have demonstrated that microscale geometric cues can control the MSC commitments, and surface micropatterns that promote contractile cytoskeleton can direct MSCs towards osteoblast lineages.

Moreover, ordered micropatterns have been also created onto CaP bioceramics [68,69,70]. For instance, our group compacted HA powders into HA disc-shaped pellets via uniaxial pressing, and polystyrene resin (PS) microspheres of varying sizes were used as pore-forming particles (poroshifters) to create a series of regular concaves (~53, 204, 508 μm in diameter) on HA substrates. Studies showed that concaves with the smallest size (~53 μm) displayed the strongest osteoinductive ability [71,72]. It is intriguing to note that the similar effects are observed in other reports. Fang et al. [73] fabricated HA ceramics of micropatterned surfaces with quadrate convexes in different sizes (~24, 55, 110 μm) by using nylon sieves as templates, showing that gene expression of osteogenic markers decreased with the increase of micropattern size. These findings have suggested that CaP ceramics with ordered surface micropatterns near cell size (20–50 μm) exert strong stimulation of cell response (e.g., promoting osteogenic differentiation).

#### 3.3.2. Nanoscale Surface Topography

From a bionic point of view, natural bone tissue has a structure with a nanoscale topography, as its main inorganic components are hydroxyapatite nanocrystals with 2–5 nm in thickness and 20–80 nm in length. Numerous studies on scaffold surface topography aim to reproduce nanoscale topography of natural bone via nanotechnology techniques [74,75,76,77]. It has demonstrated that nanotopographical features in the form of nanopits, nanoislands (nanocolumns), nanogrooves and nanotubes show a significant influence on the cell behaviors.

##### **Nanopits** 

Dably’s group [78] used electron beam lithography (EBL) and hot embossing to generate arrays of nanoscale pits (120 nm in diameter and 100 nm in depth) with different arrangements on polymethylmethacrylate (PMMA) substrate. It was found that highly ordered and random nanopit arrays displayed low osteogenic differentiation of human MSCs, while nanopits with moderate disorder increased osteoblastic differentiation as evidenced by significantly more pronounced staining of osteospecific markers–osteopontin (OPN) and osteocalcin (OCN). Besides human MSCs [78], it was also reported that the nanodisorder, rather than the highly ordered oriented patterns, could promote rapid osteogenesis from embryonic stem cells (ESCs) [79] and mature osteoblasts [80]. Previous studies also showed that nanopatterns with precise order could reduce cell adhesion [78,81,82]. These findings suggested that the arrangement (disorder degree) of nanotopographical patterns could exert significant effects on the cell responses, in particular osteogenic differentiation.

##### **Nanoislands/Nanocolumns** 

Researchers also fabricated nanoislands (13–95 nm in height) and cylindrical nanocolumns (100 nm in diameter, 160 nm in height) onto PMMA substrates via colloidal lithography, polymer demixing, and hot embossing. Previous studies found that these polymer demixed nanotopographies could play an important role in mediating the adhesion, spreading, and differentiation of various cell types, including epithelial cells [83], endothelial cells [84], fibroblasts [85,86,87,88], osteoblasts [89,90], and osteoprogenitors [91]. 13-nm-high nanoislands could induce the largest cell response as evidenced by increased initial cell adhesion, accelerated cell spreading, and a well-defined cytoskeleton, whereas the larger nanoisland (e.g., 45 and 95 nm in height) and nanocolumns resulted in the reduced cytoskeletal organization and decreased long-term cell adhesion. These findings have demonstrated that cell response may be negatively correlated with nanotopography size. Further studies offered compelling evidence that nanoislands below 50 nm (e.g., 10 and 33 nm in height) could provide crucial cues for osteoprogenitors by stimulating the differentiation of hMSCs toward an osteoblastic phenotype [92].

##### **Nanogrooves** 

Another typical form of nanotopographical pattern—nanogroove—was generated by using electron beam (EB) lithography, etching and replication molding techniques. It was found that nanogrooves could facilitate initial cell extension [93], direct cell alignment (orientation) [94], modulate cell migration along grooves [95], and mediate cell differentiation [96,97,98,99]. Fuijita et al. [94] observed the dynamic behaviors of living hMSCs on the substrate with nanogrooves (200 nm groove depth, 670 nm groove width, 870 nm ridge width) via time-lapse microscopes. It was intriguing to note that cell orientation might be attributed to the anisotropic retraction rate of cell protrusions as (⊥groove) >> (∥groove). Moreover, Abagnale et al. found that groove sizes determined the MSC commitment, as large microgrooves (15 μm) induced adipogenesis, while, small microgrooves (2 μm) stimulated osteogenesis [99]. In presence of differentiation media, nanogrooves (650 nm) promoted MSC differentiation towards both adipogenic and osteogenic lineages [99]. However, Kim et al. [98] reported that nanogrooves (350 nm) suppressed osteogenic differentiation of human dental pulp stem cells (hDPSCs) but promoted their adipogenic differentiation. Another work of their group [97] investigated the effects of the spacing ratio (i.e., ridge: groove) on cell differentiation. It was found that moderately dense (spacing ratio as 1:3) nanogrooves could enhance osteogenesis of hMSCs, which might be in connection with cell morphology and cell-substrate interaction (e.g., integrin β1 expression). These findings have indicated the possible existence of optimized nanotopographical size and density for stem cell fate. Loesberg et al. [100] pointed out that the lowest threshold of nanogroove dimensions to influence cell guidance (orientation and alignment) was around 35 nm in height. More recently, McNamara et al. [101] found that cells were sensitive to nanoscale features of 8 nm in height, suggesting that sub-10-nm nanotopographical features might also exert a significant influence on cell functions.

##### **Nanotubes** 

Metallic materials (e.g., titanium and titanium alloys) have been widely used as orthopedical implants in clinic, due to good mechanical properties. To improve their bioactivity, highly ordered nanotubular arrays with tunable sizes inferior to 100 nm are generated onto a metal surface by using electrochemical anodic oxidation (anodization), which is a low-cost, versatile and reproducible method to readily tailor nanotopographic surface for metals [102,103,104,105,106,107]. The diameter, wall thickness, and length of titania (TiO_2_) nanotubes can be precisely controlled by the condition of the anodization process, including applied potential, current density, reaction time, pH value, and electrolyte viscosity. Wilmowsky et al. [104] used an in vivo pig skull defect model to show that compared to pure titanium (Ti), implants covered with patterned 30-nm TiO_2_ nanotubes could promote bone formation by enhancing expression of type I collagen and osteocalcin (OCN). Park et al. [105,106] compared the biological functions of vertically oriented TiO_2_ nanotubes with varying diameters ranging from 15 to 100 nm. It was found that cell adhesion, proliferation, and migration increased with the decrease of nanotube diameters. The smallest nanotubes (15 nm) displaced the strongest osteogenic differentiation of rat MSCs [105] and primary human osteoblasts [106] as evidenced by the highest mineralization and osteocalcin (OCN) expression after two weeks, which might be because that 15 nm nanotubes were optimal for integrin clustering and focal contact formation to further activate down-stream signals (e.g., ERK). Brammer et al. [107] cultured mouse MC 3T3-E1 pre-osteoblasts on Ti substrates with TiO_2_ nanotubes of four different sizes (30, 50, 70 and 100 nm), finding that the small nanotubes (30 nm) increased initial cell adhesion, while, the large nanotubes (70 and 100 nm) promoted alkaline phosphatase (ALP) activity, which was the indictor of bone-forming ability.

#### 3.3.3. Potential Mechanisms for Cell Responses to Scaffold Topography

To explore its underlying mechanism, further studies demonstrated that nanotopographies (nanopit patterns) could stimulate changes in cytoskeleton arrangement [108], modulate nuclear organization that was correlated with spatially regulated gene expression [109], and enhance ability of growth hormone receptors (GH) to activate MAPK signal pathways [110].

Numerous studies have so far demonstrated that surface topographic cues of scaffolds can direct stem cells into distinct lineage differentiation, which is at least partly attributed to adhesion-related mechanisms, for cells can sense scaffold surface topography via adhesion receptors, in particular integrin molecules. Previous studies found that MSCs with constrained morphology and smaller adhesion resulted in adipogenesis, while MSCs with encouraged spread and larger adhesion were inclined to osteogenesis [65,67], suggesting that the formation of integrin-related adhesion assembly might be a critical step towards osteoblastic lineages [111]. Integrins are cell-surface trans-membrane receptors, which play an important role in the transduction of “outside-in signaling” [112,113,114]. The extracellular domains of integrins can bind to peptide ligands (e.g., RGD) that are usually derived from proteins adsorbed onto scaffold surface, and then integrin-ligand binding causes interaction and clustering of cytoplasmic proteins (e.g., focal adhesion kinase (FAK), vinculin, paxillin) to form focal adhesion complexes and trigger intracellular signaling cascades (e.g., MAPK signals) [110,115]. It is well known that once biomaterials are implanted into the host body, the first event is the rapid adsorption of plasma proteins onto the scaffold surface prior to cell arrival [116]. Scaffold properties (e.g., chemical composition, surface topography, surface charge, and surface wettability) show a significant influence on the amount, type, and configuration of the adsorptive proteins [117,118], which subsequently provide ligands for integrin bindings to further affect cell fates. Besides integrin-ligand bindings, literature also suggests that cells can directly use discrete nanolength projections–“nanopodia” to experience contact guidance from nanotopographical features and garner geometric information via nanoscale adhesion-localized structures, including adhesion-related particles [111]. Moreover, integrin-related focal adhesion also links to contractile stress fibers (i.e., cytoskeletal elements), indicating that surface topography-mediated integrin binding/clustering can induce cytoskeletal tension [108,119]. On one side, cytoskeletal reorganization may cause the stretching of the cell membrane to control the switch of ion channels, resulting in cell response [120]. On the other side, actin-based cytoskeleton can link to nuclear matrix via adaptor proteins [121], and cytoskeletal contraction may also lead to the deformation of a nuclear membrane that regulates the opening of nuclear pores to determine mRNA transportation into cytosol and further affect protein translation [120]. Recent evidence also demonstrate that cytoskeletal remodeling in response to surface topography can rapidly affect nuclear morphology (nucleoskeleton arrangement), resulting in chromosomal translocation and epigenetic DNA to activate/inactivate key genes that are involved in cell growth and functions [122].

## 4. Outlook and Perspectives

In the process of bone tissue regeneration, bone formation is largely affected by physico-chemical cues in the surrounding microenvironment. Tissue cells reside in a complex circumstance, where they can interact with each other and respond to multiple stimulus (signals) provided by the physiological microenvironment and the surrounding matrix (scaffolds). Artificial scaffolds should provide a certain circumstance full of structural cues to affect MSC differentiation, osteoblast growth, ECM deposition, and subsequent new bony tissue formation. Previous findings have helped to improve our understanding of scaffold structural microenvironmental cues that direct cell differentiation into osteoblastic lineage. Recently, advances in manufacturing techniques allow us to fabricate scaffolds with precisely designed pore structure and topographical surface in nano/micron scale, which serve as reproducible models to explore the roles of scaffold structural cues in osteogenesis. Literature has demonstrated that (1) the biomimetic nanotopographical features generally possess superior bioactivity to the micro-scale ones; (2) the disorder degree of nanopattern affects cell responses and the controlled nanodisorder favors osteogenesis; (3) the dimension and orientation of nanopatterns exert cell contact guidance to subsequently determine cell fates; (4) integrin-related focal adhesion formation and cytoskeletal reorganization are responsible for topography-induced cell functions. Therefore, scaffolds with well-designed pore structure and nanotopographical features are expected to achieve the desired cell responses. Moreover, some researchers have attempted to use easier, more efficient, and lower-costing methods (e.g., surface spraying, polishing, blasting, and etching) to increase nanoscale roughness of implant surface. All endeavors hold a promise in applying ideas of pro-osteogenic topographical cues into the practical manufacturing of artificial grafts. Future studies may focus on the interactions of cells and substrates at the atomic and molecular levels, systematically and comprehensively investigating the biofunctions of the scaffold structural cues and to decipher the underlying mechanisms. It will enlighten us to design and fabricate biomimetic orthopedic implants with proper structural features in order to provide suitable cues for inducing the differentiation of stem cells into osteoblastic lineages.

## Figures and Tables

**Figure 1 nanomaterials-08-00960-f001:**
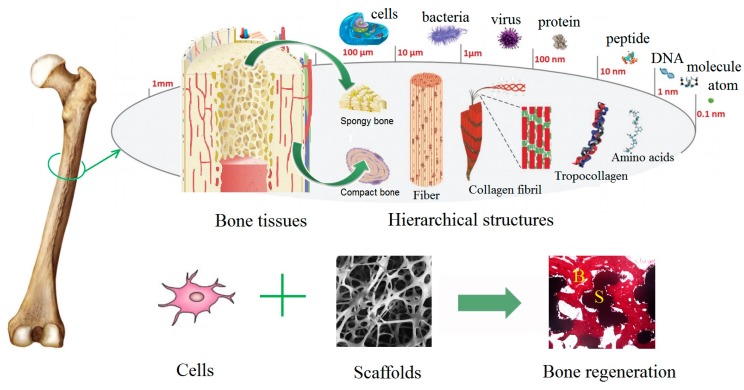
The hierarchical anatomy structures of bone tissues. Bone regeneration strategy is conducted by the synergistic effect of cells and scaffolds.

**Figure 2 nanomaterials-08-00960-f002:**
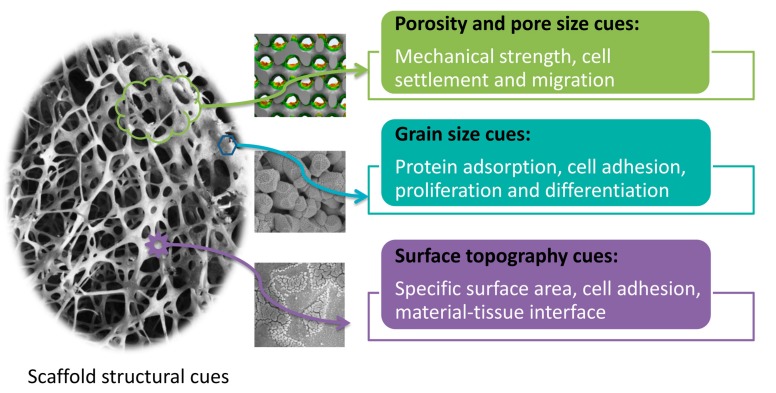
The biofunctions of scaffold structural microenvironmental cues.

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
