# Peer review of "Scaffold Structural Microenvironmental Cues to Guide Tissue Regeneration in Bone Tissue Applications"

_nanomaterials, 2018, doi:10.3390/nano8110960_

Round 1
Reviewer 1 Report
The manuscript entitled “Scaffold structural microenvironmental cues to guide tissue regeneration in bone tissue applications” by Xuening Chen et al. reviews current research on scaffold structural properties and their influence on cells. The review is well written with clearly organized topics dealing with different scaffold properties.
There are only few minor comments to the manuscript
1) It is hard to read the caption in figure 1 – bone cells, bacteria,… - they are too small, I recommend a little bit bigger size
2) In section 2. the title speaks about bone defects and bone tissue regeneration, I do not see anything concerning bone tissue regeneration in this paragraph. There are described current methods how to fill the missing bone.
3)Line 81 – The sentence “ Some researchers have reported … “ demanded a citation in my opinion.
4) Line 211 – I do not understand the citation 74 after the sentence describing hydroxyapatite crystal, which is already cited on the 1st page by citations 4 and 5.
5) Line 546-547 – The citation 71 is not complete.
Author Response
-Reviewer 1
The manuscript entitled “Scaffold structural microenvironmental cues to guide tissue regeneration in bone tissue applications” by Xuening Chen et al. reviews current research on scaffold structural properties and their influence on cells. The review is well written with clearly organized topics dealing with different scaffold properties.
There are only few minor comments to the manuscript
1) It is hard to read the caption in figure 1 – bone cells, bacteria,… - they are too small, I recommend a little bit bigger size.
Reply and Revision 1: We have revised it to bigger size.
2) In section 2. the title speaks about bone defects and bone tissue regeneration, I do not see anything concerning bone tissue regeneration in this paragraph. There are described current methods how to fill the missing bone.
Reply and Revision 2: We have introduced some content for bone regeneration. Please see the revision in red color in Section 2.
3)Line 81 – The sentence “ Some researchers have reported … “ demanded a citation in my opinion.
Reply and Revision 3: We have added the citations [14] and [15].
4) Line 211 – I do not understand the citation 74 after the sentence describing hydroxyapatite crystal, which is already cited on the 1st page by citations 4 and 5.
Reply and Revision 4: Wrong citation of ref. [74]. We have revised it.
5) Line 546-547 – The citation 71 is not complete.
Reply and Revision 5: We have revised it.

Reviewer 2 Report
very interesting review and well written. I suggest to add one small new section under the nanotopography. Layer-by-layer assembly can allow the formation of nanoscale layers for the MSCs differentiation in bone. A recent reference on this new section can be: Multilayer Nanoscale Encapsulation of Biofunctional Peptides to Enhance Bone Tissue Regeneration In Vivo. Adv Healthc Mater. 2017 Apr;6(8). doi: 10.1002/adhm.201601182.Author Response
-Reviewer 2
very interesting review and well written. I suggest to add one small new section under the nanotopography. Layer-by-layer assembly can allow the formation of nanoscale layers for the MSCs differentiation in bone. A recent reference on this new section can be.
Reply and Revision 1: We have added this reference as:
[74] P Gentile, AM Ferreira, JT Callaghan, CA Miller, J Atkinson, Multilayer nanoscale encapsulation of biofunctional peptides to enhance bone tissue regeneration in vivo. Adv Healthc Mater. 6 (2017). doi: 10.1002/adhm.201601182.
Reviewer 3 Report
This review covers various physical features of scaffold materials and how those features may impact osteogenesis and, in turn, be applied to bone repair. Overall, the review is well referenced and does an adequate job of covering the various aspects of how structural features have an impact on cell behavior. However, there are several modifications that would greatly improve this review to make it more valuable to the readership. These are listed below.
1. There needs to be a detailed review of the English usage in this manuscript. There are simply far too many awkward sentences and awkward word choices for this reviewer to address. Perhaps about 12-20 per page. They are not terrible mistakes, but they seriously detract from the message and need to be addressed.
2. There are some sections that appear to be somewhat sophomoric, such as lines 25-29. It would have been more useful to list the actual mechanical properties of bone than simply state they are "remarkable", and it is safe to assume most readers will know that the presence of bone tissue defines vertebrate animals.
3. A similar, somewhat sophomoric, description of bone defects is found in lines 55-59. Again, perhaps some genuine numerical examples would make this introductory sentence more impactful. Also, the phrase "to treat rapidly growing bone defects" has to go, but that is just one of many English usage issues.
4. The colors used for the text in Figure 2 need to be changed so that they are legible. The purple and red combination, especially, clashes too much.
5. Lines 100-117 comes across as more of a listing of different observations of pore size effects than it does an analysis of how pore size impacts cell behavior. There seems to be disagreement among different authors as to the impact of pore sizes. So what is the take home message for the readers? The inclusion of the exact results the different studies showed might shed light on the impact of each study.
6. More compliment than criticism: Section 3.2 was the first section that included sufficient experimental details to be useful to readers. This is a good thing and this same standard should be applied to the remainder of the review.
7. Lines 218-220: This reviewer is unclear about some definitions used here. How do "systemic" or "random nano-pit arrays" differ from "nano-displaced arrays"?
8. Line 223: It is difficult to imagine what "controlled disorder" is and how it might be quantified or characterized. Could the authors please clarify?
9. Lines 250-252: What is the specific evidence used here to demonstrate adipogenesis or osteogenesis?
10. Line 258: The authors discuss the "neurogenesis of MSCs", which is something that is not generally accepted by the scientific community. This highlights the need to include specific outcome details of these experiments, such as markers of neurogenesis used, which would avoid making the controversial conclusion that MSCs become nerves.
11. Line 276: It is unclear what "resist shear forces caused by implant insertion" means. Do the cells resist the shear forces? Are there even cells present? Does the material reduce shear forces? It is not clear. Also, it is not clear what "per-implant bone formation" means.
12. Line 285: This reviewer disagrees that MC 3T3-E1 cells are MSCs. 3T3-E1 cells are more of a committed osteogenic cell.
13. Line 345: "idea" is a more appropriate word than "theory".
Author Response
-Reviewer 3
This review covers various physical features of scaffold materials and how those features may impact osteogenesis and, in turn, be applied to bone repair. Overall, the review is well referenced and does an adequate job of covering the various aspects of how structural features have an impact on cell behavior. However, there are several modifications that would greatly improve this review to make it more valuable to the readership. These are listed below.
1. There needs to be a detailed review of the English usage in this manuscript. There are simply far too many awkward sentences and awkward word choices for this reviewer to address. Perhaps about 12-20 per page. They are not terrible mistakes, but they seriously detract from the message and need to be addressed.
Reply and Revision 1: We revised the whole paper and corrected the sentence and grammar. Please see the red color are the corrections.
2. There are some sections that appear to be somewhat sophomoric, such as lines 25-29. It would have been more useful to list the actual mechanical properties of bone than simply state they are "remarkable", and it is safe to assume most readers will know that the presence of bone tissue defines vertebrate animals.
Reply and Revision 2: We revised the whole paper and corrected the sentence and grammar. Please see the red color are the corrections.
3. A similar, somewhat sophomoric, description of bone defects is found in lines 55-59. Again, perhaps some genuine numerical examples would make this introductory sentence more impactful. Also, the phrase "to treat rapidly growing bone defects" has to go, but that is just one of many English usage issues.
Reply and Revision 3: We revised these sentences to avoid sophomoric description.
4. The colors used for the text in Figure 2 need to be changed so that they are legible. The purple and red combination, especially, clashes too much.
Reply and Revision 4: We revised this Figure. It is much clear now.
5. Lines 100-117 comes across as more of a listing of different observations of pore size effects than it does an analysis of how pore size impacts cell behavior. There seems to be disagreement among different authors as to the impact of pore sizes. So what is the take home message for the readers? The inclusion of the exact results the different studies showed might shed light on the impact of each study.
Reply and Revision 5: We revised this content to make it more clear.
6. More compliment than criticism: Section 3.2 was the first section that included sufficient experimental details to be useful to readers. This is a good thing and this same standard should be applied to the remainder of the review.
Reply and Revision 6: We try to provide enough research results for readers to summarize the latest findings.
7. Lines 218-220: This reviewer is unclear about some definitions used here. How do "systemic" or "random nano-pit arrays" differ from "nano-displaced arrays"?
Reply and Revision 7: We revised this content to make it more clear.
8. Line 223: It is difficult to imagine what "controlled disorder" is and how it might be quantified or characterized. Could the authors please clarify?
Reply and Revision 8: We revised this content to make it more clear.
9. Lines 250-252: What is the specific evidence used here to demonstrate adipogenesis or osteogenesis?
Reply and Revision 9: We provide corresponding references.
10. Line 258: The authors discuss the "neurogenesis of MSCs", which is something that is not generally accepted by the scientific community. This highlights the need to include specific outcome details of these experiments, such as markers of neurogenesis used, which would avoid making the controversial conclusion that MSCs become nerves.
Reply and Revision 10: We have deleted these contents to avoid any further controversial conclusion.
11. Line 276: It is unclear what "resist shear forces caused by implant insertion" means. Do the cells resist the shear forces? Are there even cells present? Does the material reduce shear forces? It is not clear. Also, it is not clear what "per-implant bone formation" means.
Reply and Revision 11: We have deleted these unclear contents.
12. Line 285: This reviewer disagrees that MC 3T3-E1 cells are MSCs. 3T3-E1 cells are more of a committed osteogenic cell.
Reply and Revision 12: We have deleted these contents to avoid any further controversial conclusion.
13. Line 345: "idea" is a more appropriate word than "theory".
Reply and Revision 13: We have corrected it. Thanks for your good comments.

Round 2
Reviewer 3 Report
The authors have addressed several of this reviewers comments, Items 3-8 and 10-13 were addressed. The biggest issue was the English usage noted in comment number 1, which was not addressed at all. There are far too many English errors for this manuscript to be acceptable. This lack of correct English usage renders many of the statements either vague or impossible to clearly understand..
. Other minor issues still remaining are, item number 2, where the change was no better than the original in that it was still too simplistic. For item number 9, the authors didn't describe the actual data, which was the question, but provided only a reference.
Author Response
Comments from the editors and reviewers:
-Reviewer 3
The authors have addressed several of this reviewers comments, Items 3-8 and 10-13 were addressed. The biggest issue was the English usage noted in comment number 1, which was not addressed at all. There are far too many English errors for this manuscript to be acceptable. This lack of correct English usage renders many of the statements either vague or impossible to clearly understand.
Reply and Revision 1: We revised the whole paper and corrected the sentences and grammars. Please see the red color are the corrections.
Other minor issues still remaining are, item number 2, where the change was no better than the original in that it was still too simplistic. For item number 9, the authors didn't describe the actual data, which was the question, but provided only a reference.
Reply and Revision 2:
item number 2. There are some sections that appear to be somewhat sophomoric, such as lines 25-29. It would have been more useful to list the actual mechanical properties of bone than simply state they are "remarkable", and it is safe to assume most readers will know that the presence of bone tissue defines vertebrate animals.
Reply and Revision for item number 2: We revised previous item 2 and provided the actual data. Please see the red color are the corrections.
tem 9. Lines 250-252: What is the specific evidence used here to demonstrate adipogenesis or osteogenesis?
Reply and Revision for item number 9: The provided reference have showed the results. Please refer to the details pictures below. Because of the pictures copyright, so this article did not cite these pictures, readers can refer to the original papers. Thanks for your good comments.
